

# Continuing the MLS water vapor record with OMPS LP

Michael D. Himes[1,2], Natalya A. Kramarova[2], Krzysztof Wargan[3,2], Sean M. Davis[4], and Glen Jaross[2]

[1]Morgan State University, Baltimore, MD, USA
[2]NASA Goddard Space Flight Center, Greenbelt, MD, USA
[3]Science Systems Association, Inc., Lanham, MD, USA
[4]NOAA Chemical Sciences Laboratory, Boulder, CO, USA

**Correspondence:** Michael D. Himes (michael.d.himes@nasa.gov)

**Abstract.** Stratospheric water vapor (SWV) plays an important role in atmospheric chemistry, dynamics, and radiative forcing. Satellite measurements by the Aura Microwave Limb Sounder (MLS), SciSat-1 Atmospheric Chemistry Experiment (ACE), and Stratospheric Aerosol and Gas Experiment III (SAGE III) on the International Space Station have provided key constraints on SWV for the past decades. MLS provides the best geographical coverage among these instruments, but it approaches the end of its life cycle in the coming years, which will result in a data desert for satellite-based SWV measurements given that ACE and SAGE III only measure at a few dozen geolocations per day. The Ozone Mapping and Profiler Suite Limb Profiler (OMPS LP) is flying aboard the Suomi National Polar-orbiting Partnership (SNPP) and NOAA-21 satellites and is planned for additional platforms in the coming years. While not designed to measure SWV, it shows weak sensitivity to it, particularly in the wake of the Hunga eruption's significant injection of water vapor into the stratosphere. By utilizing the frequent co-locations between OMPS LP and MLS measurements, we developed a neural network-based approach to retrieve SWV from SNPP OMPS LP radiances between 11.5–40.5 km. We find that the LP SWV profiles typically agree with MLS within 5%, and agreement with ACE and SAGE III is typically within 10%. We show that the SNPP-trained model is applicable to NOAA-21 OMPS LP without retraining, though minor differences in radiances between the instruments results in a ~5% bias under most conditions. Our results suggest that OMPS LP can continue the global water vapor record in the lower stratosphere into the 2030s, though continued independent measurements by satellite and balloon-borne instruments will be key to verifying the stability of our approach for quantifying decadal-scale SWV variability.

## 1 Introduction

Stratospheric water vapor (SWV) influences atmospheric dynamics, chemistry, and radiative forcing (e.g., Ramanathan and Inamdar, 2006; Charlesworth et al., 2023; Niemeier et al., 2023; Fleming et al., 2024). While SWV is typically 3–6 parts per million by volume (ppmv), water vapor concentrations in the upper troposphere can reach up to 1000 ppmv (Read et al., 2022). Deep convective systems and tropical upwelling via the Brewer-Dobson circulation can transport tropospheric air into the lower stratosphere, which comprises an important contribution to SWV (e.g., Fueglistaler et al., 2009; Khaykin et al., 2009; Randel and Jensen, 2013; Dauhut et al., 2016). Rising atmospheric temperature due to climate change increases the amount of water vapor held in tropospheric air, which in turn increases the amount of water vapor transported into the stratosphere (Yue





et al., 2019; Nowack et al., 2023). Long-term satellite measurements of SWV provide key constraints on the SWV budget and serve as important sources for data assimilation and validation of reanalysis frameworks (e.g., Davis et al., 2017; Hersbach et al., 2020; Wargan et al., 2023; Knowland et al., 2025).

Presently, satellite retrievals of SWV profiles are performed by the Aura Microwave Limb Sounder (MLS; Livesey et al., 2021), SciSat-1 Atmospheric Chemistry Experiment Fourier Transform Spectrometer (ACE-FTS; Boone et al., 2023), Strato-
spheric Aerosol and Gas Experiment III (SAGE III) aboard the International Space Station (Davis et al., 2021; Park et al., 2021), and the Thermosphere Ionosphere Mesosphere Energetics Dynamics (TIMED) satellite's Sounding of the Atmosphere using Broadband Emission Radiometry (SABER; Rong et al., 2019) instruments. These instruments provide well validated $H_2O$ products that agree among each other as well as with ground-based and in-situ measurements (e.g., Carleer et al., 2008; Hurst et al., 2014; Rong et al., 2019; Davis et al., 2021; De Los Ríos et al., 2024). Since May 2024, the MLS receiver used for
the $H_2O$ retrievals now only operates 6 days per month due to power constraints and will continue to do so until the end of the Aura mission, which significantly limits the spatiotemporal coverage of the MLS $H_2O$ product. Following the decommissioning of Aura, ACE-FTS and SAGE III will continue to provide their $H_2O$ products, but their geographical coverage is limited given they are solar occultation instruments. SABER takes around 1400 scans per day and, depending on time of year, views between 52°S–83°N or 83°S–52°N, but the local times of the measurements change by up to 12 hours over TIMED's two-month yaw
cycle. The Canadian High-altitude Aerosols, Water vapour, and Clouds mission (HAWC; Langille et al., 2025) is planned to launch early next decade, presenting a gap in global geographical coverage of SWV between Aura's decommissioning and HAWC's launch (Salawitch et al., 2025).

The Ozone Mapping and Profiler Suite (OMPS) Limb Profiler (LP) is currently flying aboard the Suomi National Polar-orbiting Partnership (SNPP; launched October 28, 2011) and NOAA-21 (launched November 10, 2022) satellites, and it is
planned for the Joint Polar Satellite System (JPSS) 4 and 3 satellites, which are estimated to launch in 2027 and 2032, respectively. Using 3 slits, the instrument measures limb-scattered radiances between 290–1000 nm with a 1 km vertical sampling. Each satellite completes 14–15 Sun-synchronous orbits per day. SNPP's OMPS LP takes measurements every ∼1° latitude, while NOAA-21's LP takes measurements every ∼0.4° latitude, resulting in around 7000 and 17,500 measured radiance profiles per day, respectively. However, OMPS LP is only weakly sensitive to $H_2O$, which has challenged the application of
traditional radiative transfer-based retrieval methods.

Our solution to retrieve water vapor profiles from OMPS LP measurements is deep learning (Goodfellow et al., 2016). Neural networks (NNs) learn to model complex, nonlinear processes in a data-driven manner. Given a set of corresponding inputs and outputs, the NN weights are tuned to approximate the underlying process, without explicit knowledge about it. Given OMPS LP's weak sensitivity to $H_2O$ and the well validated MLS $H_2O$ product, NNs could learn to accurately predict
$H_2O$ profiles from OMPS LP measurements at altitudes with sufficient sensitivity by using co-located MLS $H_2O$ profiles as the target outputs. This would result in an MLS-like $H_2O$ product, thereby continuing the MLS global SWV record following the end of the Aura mission.

Here we present an OMPS LP water vapor product between 11.5–40.5 km produced by a NN trained on co-located LP-MLS measurements. In Section 2 we investigate the sensitivity of LP to water vapor under conditions before and after the Hunga





eruption. In Section 3 we discuss our methodology to train the NN as well as validate its predictions using other instruments. We present and discuss our results in Section 4, including the limitations of our approach. Finally, we present our conclusions in Section 5.

## 2    OMPS LP sensitivity to $H_2O$

We investigate OMPS LP's $H_2O$ sensitivity using the Gauss-Seidel limb scattering radiative transfer model (RTM) of Lough-
man et al. (2004). To calculate $H_2O$ cross sections, we use the HITRAN 2020 database (Gordon et al., 2022) via the HITRAN Application Programming Interface (Kochanov et al., 2016). We then convolve those high-resolution cross sections with the OMPS LP bandpasses such that RTM calculations at a given wavelength will be more consistent with what LP would measure for the assumed conditions. Using two selected co-located MLS $H_2O$ profiles from before and after the Hunga eruption, we simulate radiances between 550–1025 nm in 5 nm intervals at altitudes of 3.5–50.5 km in 1 km intervals. For the two RTM
simulations, all parameters are kept the same except for the $H_2O$ profile to ensure that any differences in the Jacobians are due solely to differences in $H_2O$.

Based on the Jacobians output by the RTM, we select 12 wavelengths measured by OMPS LP (554, 596, 654, 720, 728, 824, 917, 929, 943, 956, 970, and 983 nm) that show the highest sensitivity to $H_2O$ in their spectral region. Figure 1 shows an example of these Jacobians at 945 nm for the selected $H_2O$ profile after the Hunga eruption. The $H_2O$ enhancement between
20–30 km attributable to the Hunga eruption results in as much as a four times increase in LP's sensitivity to $H_2O$. However, LP is very weakly sensitive to $H_2O$, and the sensitivity becomes negligible above 30 km.

## 3    Deep learning methods

### 3.1    Data curation

Since there is currently no water vapor product derived from OMPS LP measurements, we instead use the MLS version 5 water
vapor product as our ground truth. To prioritize times where SNPP and Aura have closely aligned orbits, we select dates that have

–    at least one orbit with 60 consecutive co-locations that are within 30 minutes and within 100 km, and

–    at least 250 total co-locations on that day that satisfy the above co-location criteria.

These criteria are satisfied every couple days due to their similar equatorial crossing times around 1:30 in the afternoon. On
dates between February 2014 and December 2024 that satisfy the above criteria, we co-locate OMPS LP and MLS measurements within 6 hours and 100 km to build a data set of OMPS LP radiances and the corresponding MLS water vapor profiles. This results in 2,074,101 co-locations, with almost half occurring at high latitudes. For context, SNPP OMPS LP collects over 2.5 million measurements per year.





**Figure 1.** Example of OMPS LP's sensitivity to $H_2O$ at 945 nm in the tropics. Jacobians for selected MLS $H_2O$ profiles **(a)** before and **(b)** after the Hunga eruption peak in the upper troposphere due to the increased water vapor content. Panel **(c)** shows the selected water vapor profiles, which show an enhancement in water vapor around 24 km attributable to the Hunga eruption. Panel **(d)** shows the ratio of (b) to (a). The increased water vapor concentration between 21–27 km due to Hunga results in LP's sensitivity increasing by up to 4× in this altitude range.



We limit the MLS water vapor profiles to $\leq 261$ hPa, as the 316 hPa pressure level can be affected by the a priori profiles used in the MLS retrievals. We log-linearly interpolate the water vapor profiles from the MLS pressure grid to OMPS LP's geometric height (11.5–40.5 km in 1 km steps) using the NASA Global Earth Observing System Forward Processing for Instrument Teams (GEOS FP-IT; Lucchesi, 2015) pressures. We limit the altitude range to 11.5–40.5 km because 10.5 km can exceed 261 hPa and the $H_2O$ sensitivity of OMPS LP becomes $\sim 0$ above 40.5 km.

We also consider a similar methodology but for ACE and SAGE III data using co-location criteria of within 1 day, within 2° latitude, and within 1113 km longitude (equal to 10° longitude at the equator), consistent with the criteria used in Davis et al. (2021). These data sets are used to investigate whether it is viable to train exclusively on ACE or SAGE III data and whether training on a combination of MLS, ACE, and/or SAGE III data offers benefits over only training on MLS data.

## 3.2 Neural network methods

For each co-located measurement, we construct an input–output pair to be used during NN training. The inputs are comprised of

- LP radiances at 554, 596, 654, 720, 728, 824, 917, 929, 943, 956, 970, and 983 nm,

- FP-IT pressures and temperatures, and

- the solar zenith angle of the LP measurement.

These inputs are formatted as 2-D "images" (wavelength × altitude) with four channels (radiance, pressure, temperature, solar zenith angle), similar to a standard RGBA image. The radiances vary at each point in the 2-D image, the pressures and temperatures vary only with respect to altitude, and the solar zenith angle is constant throughout. For each input image, the corresponding outputs are the co-located MLS $H_2O$ profile that has been interpolated to the LP altitude grid.

To address the latitudinal sampling bias inherent in the co-located data set, we first select a subset of the data such that there are roughly the same number of samples in each 5° latitudinal bin, resulting in 1,137,100 input–output pairs. We ensure that extrema for each latitudinal bin are included in this subset. We then split these data into training (used to update NN weights), validation (monitors for overfitting during training), and testing (tests model generality on unseen data after training is complete) sets in a proportion of roughly 75%, 15%, and 10%, respectively.

For data pre-processing and NN training, we utilize the open-source Python package MARGE (Himes et al., 2022), which uses TensorFlow (Abadi et al., 2016) via the Keras API. We pre-process the data by taking the base-10 logarithm of the OMPS LP radiances, GEOS FP-IT pressures, and MLS $H_2O$ profiles, then scale each input and output parameter to be within the closed interval [-1, 1] based on their training set extrema at each altitude.

To determine a neural network architecture well suited to solving this problem, we perform a Bayesian hyperparameter optimization (Akiba et al., 2019). The selected architecture is similar to the landmark AlexNet architecture (Krizhevsky et al., 2012); for details on our optimization procedure, the selected architecture, and the training details, see Appendix A. Using the chosen architecture, we train an ensemble of 10 neural networks using a mean-squared-error loss function. The ensemble's





mean prediction provides the retrieved $H_2O$ profile, while the standard deviation among the ensemble provides an uncertainty estimate. The ensemble size was determined by adding ensemble members until the mean and standard deviation of the ensemble's predictions did not significantly change.

Additionally, we apply these methods to permutations of the co-located MLS, ACE, and/or SAGE III data sets. When
combining more than one data set, we consider two approaches, one where we use the data as is, and another where we bias correct the ACE and SAGE III data such that they have a global median difference of 0% at all altitudes with respect to MLS.

### 3.3 Evaluation and validation

To evaluate the NN's typical accuracy and how well it generalizes to unseen data, we calculate the root mean square error (RMSE) and coefficient of determination ($R^2$) for the validation and test sets. We validate our LP water vapor product by
comparing with satellite measurements, balloon-borne measurements, and an assimilation/reanalysis product.

For satellite measurements, we consider the MLS version 5 (Livesey et al., 2021), SAGE III version 6 (NASA/LARC/S-D/ASDC, 2025), and ACE version 5.3 (Boone et al., 2023) products. For MLS, we consider co-locations within 6 hours, while for SAGE and ACE we consider co-locations within 24 hours. For all three, we only consider the co-location if it is separated by less than 1000 km and within 2° latitude. When multiple co-locations satisfy these criteria, we use the co-location with the
shortest distance. For each of these instruments, we compute both a global median percent difference as well as zonal median percent differences in 5° latitude bins. We found anomalous values in the SAGE and ACE data sets that differ by a factor of up to 10,000 compared with the layers above and below it, even after applying each product's recommended screening criteria. To screen out the most extreme of these unrealistic values, we apply a very conservative $20\sigma$ median rejection routine to the data set of percent differences.

Given the reported instrumental drift in version 5 of the MLS water vapor product (Livesey et al., 2021), we investigate whether our product shows similar properties as the MLS product by performing a multiple linear regression (MLR) on monthly means for a 5°×5° grid, with proxies for a linear trend, seasonal cycle, quasi-biennial oscillation (QBO), and El Niño Southern Oscillation (ENSO), as these terms explain the majority of stratospheric $H_2O$ variability. For the seasonal cycle term, we also include a phase offset term for lag in months, given that it takes time for the effects to propagate upwards through the
stratosphere. For the ENSO term, we regress using the sea surface temperature anomaly with a fitted lag in months, as previous work showed this is necessary to maximize correlation (e.g., Garcia et al., 2007; Calvo et al., 2010; Yu et al., 2022). For the QBO term, we use coefficients for the two leading empirical orthogonal functions for the QBO wind time series between January 1956–February 2025.

To compare with balloon-borne measurements, we consider NOAA Frost Point Hygrometer (Hurst et al., 2011) and Cryo-
genic Frost point Hygrometer (Vömel et al., 2007a, b) soundings from Boulder, USA; Hilo, USA; Lauder, New Zealand; San José, Costa Rica; Lindenburg, Germany; and Biak, Indonesia. For each sounding, we co-locate satellite measurements within 24 hours, 2° latitude, and 1113 km longitude (equal to 10° at the equator). If multiple co-locations meet these criteria, we select the profile that minimizes the distance from the sounding. We calculate the median absolute difference and percent difference over the data set of co-locations between each satellite instrument and the frost point measurements.





We additionally compare with the Modern-Era Retrospective analysis for Research and Applications version 2 (MERRA-2) Stratospheric Composition Reanalysis of Aura MLS (M2-SCREAM) reanalysis product (Wargan et al., 2023), which assimilates MLS products including $H_2O$, as this guarantees a co-location for all OMPS LP measurements. To assess whether our methodology blindly memorizes the days it sees during training (where co-locations with MLS are frequent) or if it learns to generalize to days it does not see during training (where co-locations with MLS are less frequent), we compute the mean

differences and standard deviation of the differences between the LP product and M2-SCREAM for two subsets of 2021 data: one that contains the days where training data were drawn from, and another that contains the days where no data were used during training.

Finally, given that OMPS LP is onboard the NOAA-21 satellite and planned to launch onboard two additional satellites in the coming years, we apply our SNPP-trained model to NOAA-21 OMPS LP measurements to determine whether our model

can generalize to future iterations of the same instrument, as Himes et al. (2025b) found this to be the case for OMPS LP aerosol retrievals.

## 4    Results and Discussion

### 4.1    Training on MLS data

Figure 2 shows the mean and standard deviation of the $R^2$ and RMSE metrics for the NN ensemble when applied to the test

set of data not seen during training. $R^2$ is >0.7 except between 15.5–18.5 km. These altitudes probe the upper troposphere or lower stratosphere depending on latitude, and the LP product shows greater errors when compared to MLS in the troposphere. The 15.5–16.5 km altitudes, which are consistently within the troposphere in the tropics, feature the lowest $R^2$ values, while 17.5–18.5 km, which are typically not within the troposphere in the tropics, yield an $R^2$ just below 0.7. When considering only events outside the tropics, $R^2$ for 15.5–16.5 km increases to ∼0.7, while considering only events within the tropics results in

$R^2$ reducing to ∼0.55 for these altitudes. In the stratosphere, the RMSE is <10% of the $H_2O$ VMR. Errors increase below 18.5 km as measurements increasingly occur in the troposphere where water vapor VMR increases substantially. The discontinuity in RMSE at 32.5 km is related to the discontinuity in MLS v5 a priori profiles (Millán et al., 2024).

We find that omitting specific years during training can be important for certain situations. When omitting 2015–2016, we find that the model is generally unaffected and still performs well during those years. However, when omitting 2024–present,

we find that the model begins producing severely inaccurate predictions by March 2024. This behavior is likely explained by the difference between these considered periods: while 2015–2016 were ordinary years in terms of SWV, the continued presence of elevated SWV from the Hunga eruption into 2024 is atypical and not represented by the data available during training. By including a small fraction of 2024 data during training, we find that the model continues performing accurately up to the present time of writing this manuscript. As the stratosphere returns to pre-Hunga conditions, we expect that the NNs will

continue producing accurate retrievals of $H_2O$, but continued comparisons with other instruments designed to measure water vapor, such as ACE, will be critical to ensuring that accuracy.

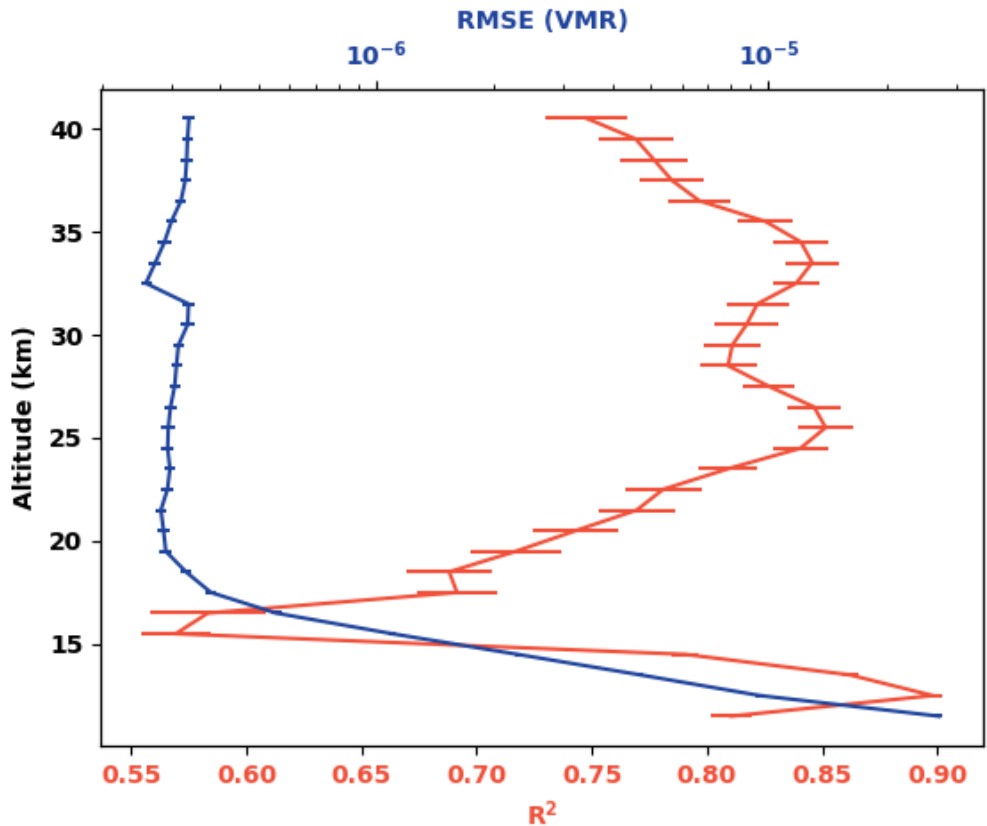

**Figure 2.** Performance summary for the ensemble of neural networks when applied to the test set. Error bars denote the standard deviation among the NNs' performances.

Since the input radiances are at wavelengths affected by aerosols, we analyze the model errors as a function of aerosol extinction reported in the OMPS LP aerosol product. We note that a weak anti-correlation exists between the predicted $H_2O$ VMR and the aerosol extinction at 675 nm in the lower stratosphere, though this is a real phenomenon rather than an artifact

of our model. Stratospheric aerosol extinction generally peaks immediately above the tropopause and decreases over the few kilometers above it, while stratospheric $H_2O$ VMR typically is at a minimum immediately above the tropopause due to the cold trap and increases over the few kilometers above it. We find that the NNs' percentage error as a function of aerosol extinction at 675 nm is uncorrelated, indicating that the success of our approach is not dependent on aerosol conditions.

In general, the error with respect to MLS is independent of the presence of tropospheric clouds. However, events affected by

polar stratospheric clouds (PSCs) show a median bias around -2% at most altitudes. When considering the error as a function of distance below the PSC, the median bias can exceed -20% at 17 km below the PSC, which only occurs for PSCs at $\geq$28.5 km. However, the standard deviation of these errors can be substantial, where it averages around 33% between 15.5–22.5 km,



with a maximum of 66% at 22.5 km. Given this behavior, we recommend that events contaminated by PSCs should not be used for scientific studies; the data product includes quality flags for these events.

## 4.2 Training on ACE and SAGE III data

We also considered training on co-located ACE and/or SAGE III data as an alternative to the co-located MLS data set. Results for these experiments were negative.

When training exclusively on ACE or SAGE III data, we find that the resulting NN models do not properly generalize to unseen data, indicating that these data sets are not sufficient to solve this problem. Part of this may be explained by the typically significant measurement time differences between LP and the other instruments; LP measures in the early afternoon, while ACE and SAGE measure at sunrise/sunset, and these times only coincide for select geolocations depending on the time of year. However, the number of co-locations seems to be a more limiting factor. When restricting the MLS data set to similar sizes as the ACE and SAGE data sets, we find that the resulting performance is poor. Given the success when using the MLS-LP data set of ∼1 million co-locations but the failure when considering tens of thousands of co-locations, these results emphasize that our methodology relies on a large data set of co-located profiles.

Additionally, when including ACE and/or SAGE III data alongside the MLS data, we find significantly degraded performance, regardless of whether or not the ACE and SAGE data were de-biased with respect to MLS. This is likely attributable to the variances between MLS, ACE, and SAGE being on the order of the natural variability of water vapor, which inhibits NN learning.

Despite our negative results when training on ACE and/or SAGE data, we cannot rule out that alternative ML approaches not considered here could utilize ACE and/or SAGE data to derive water vapor profiles from LP radiances.

## 4.3 Comparisons with satellite measurements

Figure 3 summarizes the global median percent differences between LP stratospheric $H_2O$ profiles and co-located MLS, ACE, and SAGE profiles. For this comparison, we filter all LP tropospheric measurements by using the nearest co-located tropopause altitude reported in the GEOS FP-IT product. The error bars show the standard error of the median, which is generally negligible except at low altitudes for ACE and at high altitudes for SAGE. Where LP detects a cloud, we exclude any measurements at or below the cloud top, though we find that this criterion does not significantly alter the results. Note that for the MLS comparisons we include all co-located data, including those used during training; this choice does not bias the results, as discussed later in Section 4.5.

Differences with respect to MLS are less than 2% at all altitudes ≥14.5 km, with a maximum difference of 4.1% at 11.5 km. When considering only 2025 data, the most extreme difference is 7.7% at 13.5 km, with a typical difference of ∼5% below 22 km and <2% above 25 km. Given that we trained on MLS data, this close agreement is expected and shows the model has learned a good approximation to retrieve MLS-like water vapor profiles.

When comparing with ACE, we find agreement within 10% except between 11.5–13.5 km, where differences can reach up to 19.3%. In general, the differences increase with altitude from -19.3% at 11.5 km up to 8.8% at 40.5 km. This behavior is




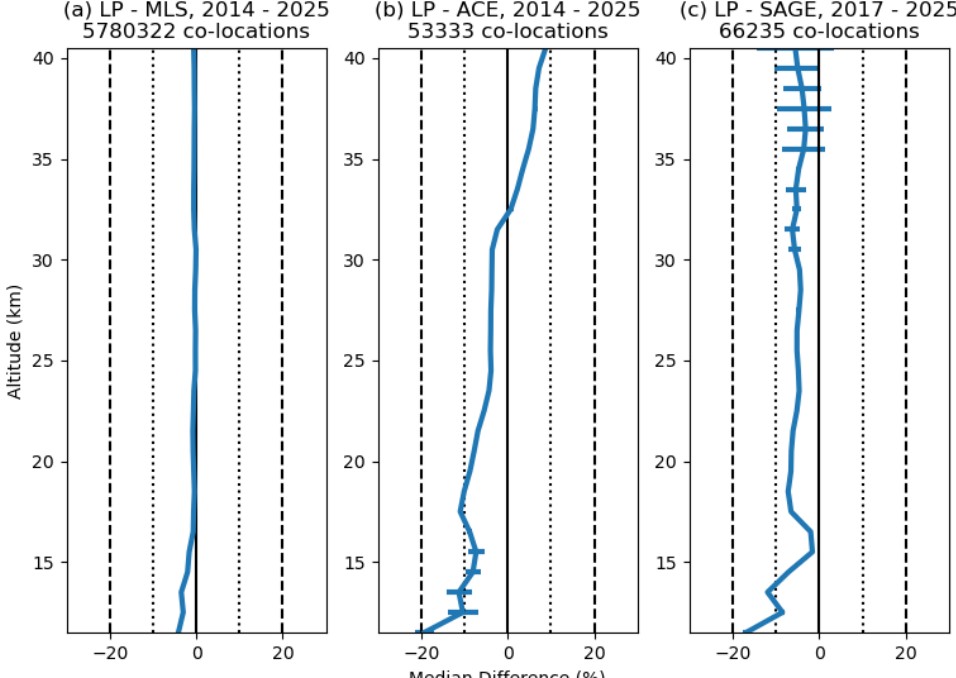

**Figure 3.** Summary of global median percent differences between the LP stratospheric H$_2$O profiles and co-located **(a)** MLS, **(b)** ACE, and **(c)** SAGE profiles. Horizontal uncertainty bars indicate the standard error of the median.

generally consistent with earlier studies, such as Davis et al. (2021) which shows a similar pattern of increasing differences between 15–40 km when comparing SAGE and ACE.

LP's differences with respect to SAGE are generally around 6% or less. Between 11.5–13.5 km, differences can reach up to 16.9%. Given that a similar increase in differences at these altitudes is seen when comparing with both ACE and SAGE but not with MLS, this suggests that either ACE and SAGE are biased high in this regime, or MLS is biased low in this regime and our LP product has inherited this bias.

Figure 4 and Figure 5 show the median percent differences and standard error of the median, respectively, in 5° latitudinal bins for the comparisons with MLS, ACE, and SAGE. The results are generally consistent with those shown in Figure 3. The notable exceptions occur at the lowest altitudes. For the comparisons with MLS, differences can reach up to 8% between 11.5 - 13.5 km just outside of the tropics. For ACE, the deviations at these altitudes can exceed 21%, but notably the standard error of the median is typically ~11% in this region, suggesting that this low bias may not be as substantial as it appears. However, comparisons with SAGE also show this low bias at these altitudes, where the standard error is negligible. Together, this suggests that the LP product has a slight systematic low bias at 11.5–13.5 km just outside the tropics, but at latitudes



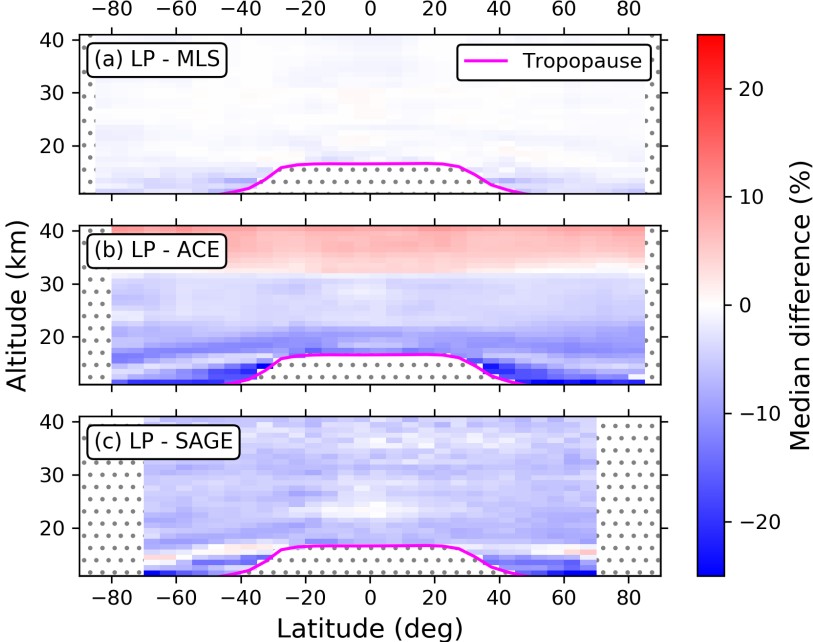

**Figure 4.** Summary of median percent differences between the LP stratospheric $H_2O$ profiles and co-located **(a)** MLS, **(b)** ACE, and **(c)** SAGE profiles in 5° latitudinal bins. The pink line indicates the median tropopause altitude for a given latitudinal bin. Gray stippling indicates where there are no data, whether due to lack of statistical significance (high latitudes) or due to being in the troposphere (below the tropopause).

$\geq$45°, the agreement with MLS suggests the biases when comparing with ACE and SAGE are related to statistical differences
between those products and MLS.

Figure 6 shows the "tape recorder" of alternating positive and negative anomalies in $H_2O$ VMR, primarily attributable to seasonal changes in $H_2O$. For this plot, we subtract the pre-Hunga mean profile from each daily zonal mean to produce the daily anomaly. Before 2025, the OMPS LP and MLS tape recorders show excellent agreement throughout the stratosphere, with OMPS LP correctly capturing the increase in $H_2O$ due to the Hunga eruption. Beginning in early 2025, OMPS LP shows
a positive bias >1 ppm above 30 km that is not seen in the corresponding MLS data. This bias is likely due to the weak $H_2O$ sensitivity at these altitudes, which inhibit the NNs' ability to reliably infer the $H_2O$ VMR at these altitudes. Since the NNs correctly infer the $H_2O$ at these altitudes before 2025, it suggests that the pre-2025 data were successfully predicted based on the shape of profiles at the lower altitudes that have sensitivity. With an absence of 2025 data in training, they guess based on similar profiles from the training set, which are evidently those influenced by the elevated $H_2O$ from Hunga. In 2025, the NNs
perform reasonably well below 30 km, indicating that there is sufficient sensitivity for the determined approximation to remain accurate when applied to unseen data, though there is a slight overestimation ($\sim$0.25 ppm) in mid-2025 between 25–30 km. We therefore advise that users exercise caution when using the OMPS LP $H_2O$ product above 30 km.



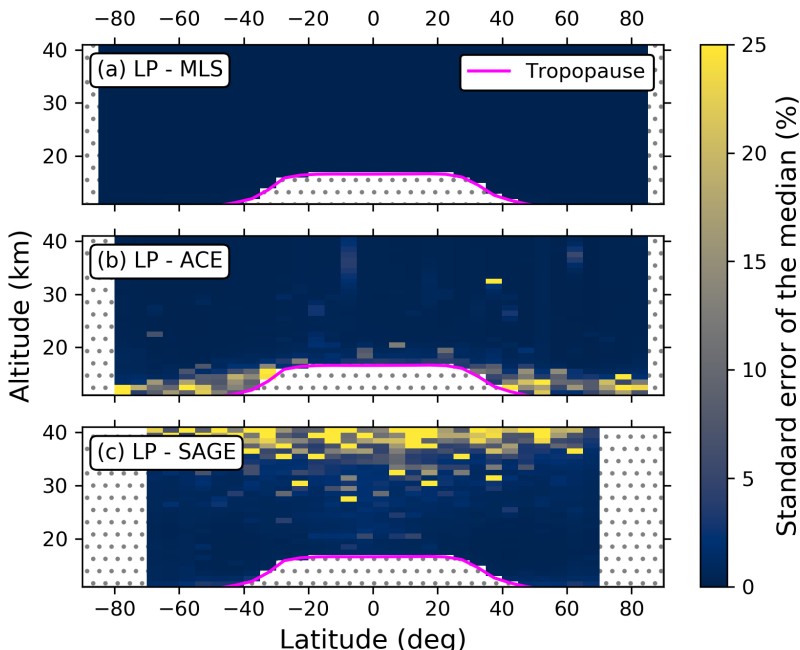

**Figure 5.** Like Figure 4, but for the standard error of the median.

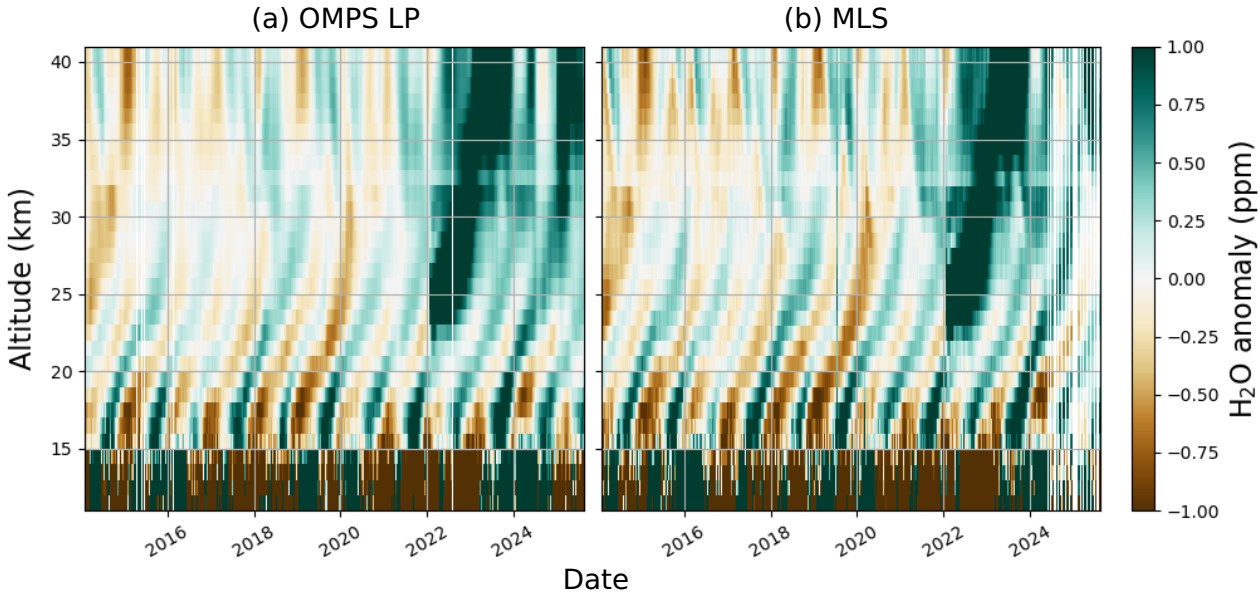

**Figure 6.** Parts-per-million anomaly in $H_2O$ VMRs for the daily zonal means within 2.5° of the Equator for the **(a)** OMPS LP and **(b)** MLS water vapor products. The anomaly is determined by subtracting the pre-Hunga mean profile from each daily zonal mean profile. Beginning in May 2024, the MLS data become more sparse due to only taking measurements 6 days each month.





## 4.4 Multiple linear regression analysis

In general, the results of our MLR analysis show similar behavior between the LP and MLS products, but the fitted coefficients
for the LP data tend to be less than the corresponding MLS coefficients. The main exception to this is the seasonal phase offset,
where both products closely agree. Figure 7 shows an example of the fitted coefficients for the seasonal amplitude at 14.5
km. The South Asian monsoon stands out clearly in both panels, though the LP product's fitted amplitudes for this region are
around 1 ppm less than the corresponding fits for MLS.

Regarding water vapor trends, the LP product generally shows greater trends in the troposphere and weaker trends in the
stratosphere when compared with MLS. Where both products show a trend of increasing $H_2O$, LP tends to show a lesser trend
than MLS (Figure 8). As MLS v5 is known to still contain some statistically significant drifts in the lower stratosphere when
compared with balloon measurements (Livesey et al., 2021), our results suggest that the NN methodology reduces these drifts.

## 4.5 Comparisons with M2-SCREAM

Figure 9 compares the OMPS LP and M2-SCREAM $H_2O$ products for the year 2021. Note that M2-SCREAM assimilates
MLS v4.2, which is biased high for water vapor, while LP is trained on MLS v5, resulting in a persistent ∼0.5 ppmv bias
between the products. Days in which data were (Fig. 9a) or were not (Fig. 9b) included during training are shown separately
but look almost identical, highlighting that the NNs' predictions are equally accurate whether or not they saw data from that day
during training. Additionally, the standard deviation of the differences between OMPS LP and M2-SCREAM are consistently
less than the standard deviation among OMPS LP or M2-SCREAM profiles, indicating that the OMPS LP product is more
accurate than the natural variability of $H_2O$. Overall, our results suggest that the NN predictions are in good agreement with
M2-SCREAM for data not seen during training.

## 4.6 Comparisons with balloon-borne measurements

Figure 10 shows the median differences between satellite instruments (MLS, SAGE, ACE, and OMPS LP) and the frost
point hygrometer soundings from the six stations considered (see Section 3.3). The LP product agrees with the frost point
measurements within 0.3 ppmv and within 10% between 16.5–27.5 km; this is in close agreement with the MLS results, which
is expected given that we trained on MLS data. The only notable difference in agreement between MLS and LP is that the LP
product shows a slightly reduced bias between 16.5–21.5 km. Like in Davis et al. (2021), we find that the satellite instruments
show a dry bias in the upper troposphere compared to the frost point measurements, which may be due to spatiotemporal
variability between the co-located measurements and/or reduced data quality in this regime.

## 4.7 Application to NOAA-21 OMPS LP

Paralleling the SNPP comparisons in Section 4.3, Figures 11 and 12 show respectively the global median percent differences
and the 5° zonal median percent differences between NOAA-21 OMPS LP and MLS, ACE, and SAGE III. For global compar-
isons, NOAA-21 OMPS LP shows a persistent ∼5% offset with respect to the corresponding SNPP comparisons at all altitudes.







**Figure 7.** Results for the seasonal amplitude at 14.5 km fitted via MLR for the **(a)** LP and **(b)** MLS water vapor products. The large amplitude over South Asia is attributable to the annual monsoon's strong seasonal impact on water vapor.







**Figure 8.** Results for the linear trend in $H_2O$ at 18.5 km fitted via MLR for the **(a)** LP and **(b)** MLS water vapor products. In general, LP shows a weaker trend of increasing $H_2O$ than MLS.





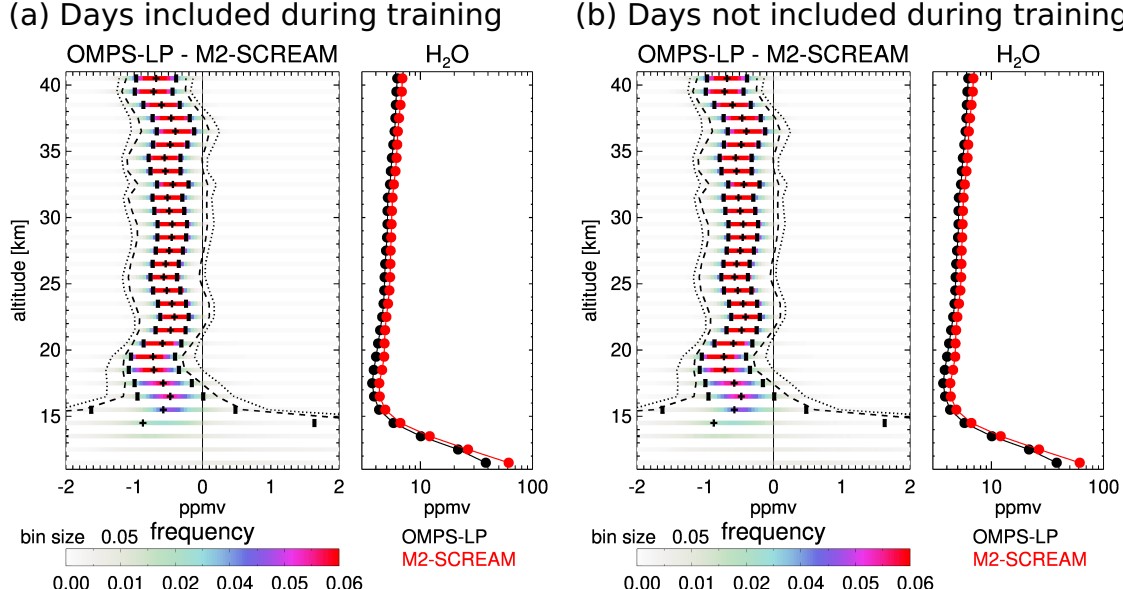

**Figure 9.** Comparison between OMPS LP and M2-SCREAM for days in 2021 where **(a)** some data were included during training and **(b)** no data were included during training. Note that for days in which data were included during training, only a small percentage (1–4%) of data on those days were used in the training data set. For each panel, the left subpanel shows the probability density function of the differences between OMPS LP and M2-SCREAM as horizontal colored bars, one standard deviation of the differences as the vertical marks, the mean difference as plus signs, the mean difference $\pm$ the standard deviation of OMPS LP $H_2O$ as dashed lines, and the mean difference $\pm$ the standard deviation of M2-SCREAM as dotted lines. The right subpanel shows the mean $H_2O$ profile for each product. Panels **(a)** and **(b)** look nearly identical, indicating that the model is retrieving $H_2O$ from the information content embedded in LP radiances rather than blindly memorizing the training data. The $\sim$0.5 ppmv offset between the products is due to differences in the MLS version used by each product.

A similar low bias is seen in the NOAA-21 OMPS LP aerosol data, suggesting that this bias is attributable to differences in radiances between OMPS LP on SNPP and NOAA-21. Given our methodology, it is unclear whether the NNs have learned to implicitly account for a bias in the SNPP radiances or if the problem is related to the calibration of NOAA-21 radiances. However, this bias is not strictly a -5% shift for all conditions; the zonal comparisons show that the tropics exhibit a positive bias not seen in the corresponding SNPP comparisons. Further investigation is necessary to understand the cause of these biases. If the origin of these biases is not able to be determined, they could be addressed via a soft calibration approach.

## 5 Conclusions

We presented a water vapor retrieval product derived from SNPP OMPS LP measurements via a neural network (NN) trained on co-located MLS version 5 water vapor profiles. In general, the LP $H_2O$ product is consistent with other water vapor products




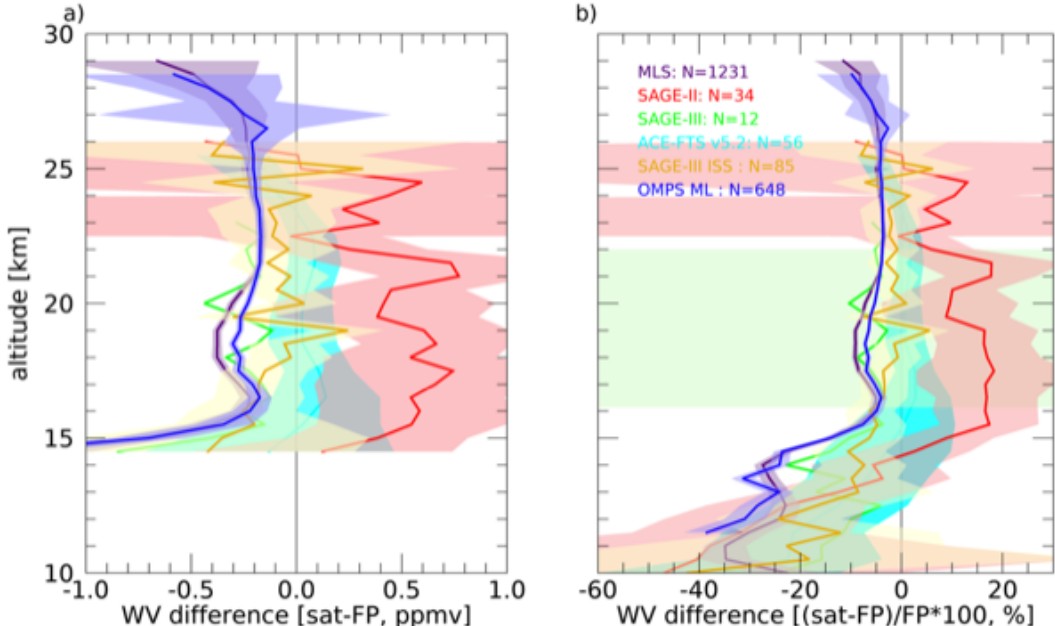

**Figure 10.** Comparisons between satellite instruments and frost point balloon measurements. Panel **(a)** shows the median difference in ppmv limited to altitudes where the mean water vapor is <10 ppmv, and panel **(b)** shows the median percent difference. In both panels, the shaded regions show the median $\pm$ 2 standard errors of the mean.

considered here. We find that our method typically agrees with MLS within 5% at all altitudes considered. The results of our multiple linear regression analysis show good correspondence between LP and MLS for seasonal water vapor variations, including for the south Asian monsoon. LP's tape recorder in the tropics also shows close agreement with MLS, capturing both the alternating positive and negative seasonal anomalies as well as the large water vapor injection from the Hunga eruption. Agreement with SAGE III version 6 and ACE version 5.3 water vapor profiles is typically within 10% above 15 km and within 20% below 15 km. When compared with frost point balloon measurements, OMPS LP generally agrees within 10% in the stratosphere, closely mirroring comparisons between those frost point measurements and MLS. Comparisons with the M2-SCREAM reanalysis product show similar behavior between days included in training and days omitted from training, indicating that our method is retrieving $H_2O$ from the LP radiances rather than memorizing the training data. Overall, we find that the LP product performs comparably to MLS over the 11.5–40.5 km altitude range considered, enabling the continuation of the MLS water vapor record for these altitudes.

When applying the same methodology but using SAGE III and/or ACE data for training, we find significantly reduced performance. We similarly find poor performance when limiting the MLS-LP data set to the same size as the SAGE and ACE data sets, which suggests that the success of our approach relies on a large training data set of co-located profiles. When including SAGE III and/or ACE data alongside MLS data, we also find poor performance, regardless of whether or not the



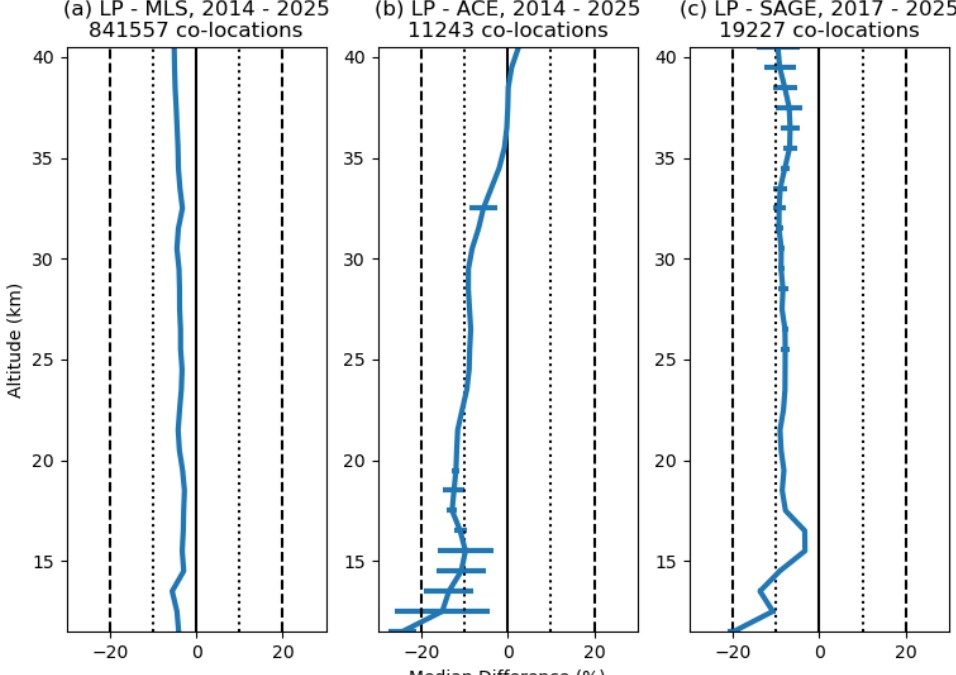

**Figure 11.** Like Figure 3, but for NOAA-21 OMPS LP.

SAGE III and ACE data are bias corrected to have a 0% median difference with MLS at all altitudes. This suggests that the variances between the three satellite data products inhibit NN learning.

Despite insufficient co-located data to train a well generalized model specific for NOAA-21 OMPS LP, we find that the SNPP-trained NN is applicable to NOAA-21 OMPS LP measurements without retraining. For NOAA-21 data, we find a persistent negative bias of ∼5% under most conditions when compared with the corresponding SNPP results; this pattern is also seen in comparisons between the SNPP and NOAA-21 OMPS LP aerosol products, suggesting that it is due to differences in the radiances rather than poor generalization of the SNPP-trained NN. However, the source of this bias is unclear at the time

of writing; future work should explore approaches to identify the origin of this bias, characterize it, and correct it, if possible. Assuming that the NN model continues performing well over the coming years, our results suggest that this SNPP-trained model will be applicable to OMPS LP onboard JPSS-4 and 3, which are planned to launch in 2027 and 2032, respectively, thereby extending the MLS water vapor record into the 2030s, albeit at a reduced altitude range. Continued satellite and balloon-borne measurements from instruments with physics-based stratospheric $H_2O$ products, such as ACE, SAGE III, and

frost point hygrometer soundings, will be integral to ensuring that our NN-based retrievals continue to perform well in the coming years.



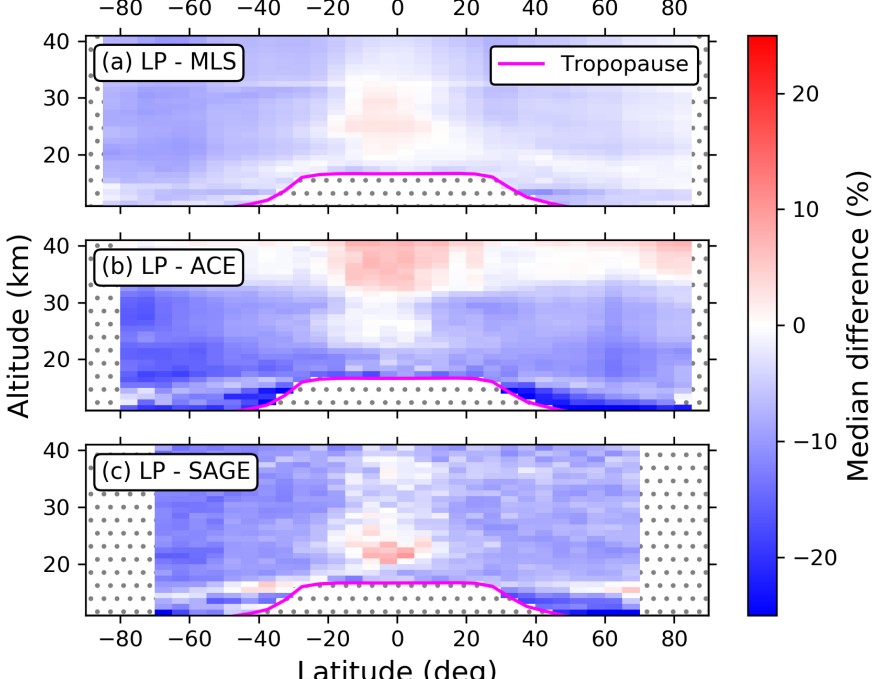

**Figure 12.** Like Figure 4, but for NOAA-21 OMPS LP.

*Code and data availability.* The MARGE software is available on GitHub at https://github.com/exosports/MARGE (Himes, 2022). All data and results related to the MARGE software for this work are publicly available under the Reproducible Research Software License at https://doi.org/10.5281/zenodo.17237404 (Himes et al., 2025a). The SNPP OMPS LP version 1.0 $H_2O$ data product is available at https://doi.org/10.5067/C1BD8BLEBH04 (Himes, 2025a). The NOAA-21 OMPS LP version 1.0 $H_2O$ data product is available at https://doi.org/10.5067/XNK38X2VQGZ0 (Himes, 2025b). The SNPP OMPS LP version 2.6 L1G data product is available at https://doi.org/10.5067/YVE3FSNJ59RQ (Jaross, 2023).

## Appendix A: Neural network optimization, architecture, and training

To optimize the neural network architecture for this problem, we performed a Bayesian hyperparameter optimization over the number and types of layers, number of nodes per layer, and activation functions. We considered fully connected NNs, convolutional NNs, and architectures that utilize both fully connected and convolutional layers. In addition to standard fully connected layers, we also considered Concrete Dropout layers (Gal et al., 2017), which include a trainable parameter for the layer's dropout rate. For convolutional architectures, we considered architectures with and without pooling layers.

The selected architecture is similar to the landmark AlexNet architecture (Krizhevsky et al., 2012), with hidden layers and activation functions consisting of Conv2D(32)–ReLU–MaxPool2D–Conv2D(64)–ReLU–MaxPool2D–CD(256)–ReLU–CD(256)–ReLU, where Conv2D($m$) indicates a two-dimensional convolutional layer with $m$ feature maps using a kernel





size of 5, ReLU indicates the rectified linear unit activation function, MaxPool2D indicates a two-dimensional pooling layer that selects the maximum value within a 2×2 window, and CD($n$) indicates a Concrete Dropout layer with $n$ nodes. This is followed by a fully-connected output layer of 30 nodes, corresponding to the $H_2O$ VMR at the 30 altitudes spanning 11.5–40.5 km. Note that other architectures performed similarly to the selected architecture; we found that the training data set played a more significant role in model performance.

We optimized the learning rate policy according to the method described by Himes et al. (2025b) and trained each NN using the mean-squared-error loss over the validation set until early stopping engaged after a patience of 60 epochs. On average, models trained for 638 epochs, which required an average of almost 7 hours to train using an Nvidia V100 graphics processing unit.

On our processing system, running our retrieval algorithm using the central processing unit requires around 12 and 16 seconds to process one SNPP and N21 orbit, respectively; specialized graphics processing units would reduce this runtime.

*Author contributions.* MDH curated the data set used for the NNs; trained the NNs; developed the software that produces the data product; performed comparisons with MLS, SAGE III, and ACE; applied MLR to separate trend, seasonal, QBO, and ENSO contributions; and wrote the initial draft of the manuscript. NAK oversaw the project, provided guidance on the validation efforts, and provided advice on figures. KW performed the comparisons between the LP $H_2O$ product and M2-SCREAM. SMD performed the comparisons between the LP $H_2O$ product and balloon-borne measurements. GJ suggested the initial idea for the project and provided advice based on previous efforts to retrieve $H_2O$ from OMPS LP. All co-authors advised ways to investigate and improve the performance of the method. NAK, KW, and SMD reviewed the manuscript and provided advice on the text.

*Competing interests.* At least one of the authors is a member of the editorial board of *Atmospheric Measurement Techniques*.

*Acknowledgements.* The authors would like to thank the OMPS LP characterization team for producing the Level-1 gridded data used in this work. We are grateful for Alyn Lambert, William Read, Luis Millan, Nathaniel Livesey, and all members of the Aura MLS water vapor team for their work on the valuable data product that made this work possible. We also thank the science teams responsible for the ACE, SAGE, and frost point hygrometer data used for validation in this work. Additionally, we thank James Johnson and Daniel Kahn for feedback on the data product formatting and assisting with releasing the data product. We also recognize contributors to NumPy, SciPy, Matplotlib, TensorFlow, Keras, Optuna, Dask, the Python Programming Language, and the free and open-source community.

*Financial support.* This work was supported by the NASA Goddard Space Flight Center Earth Science Division's Strategic Science program.



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
