# Peer review of "Continuing the MLS water vapor record with OMPS LP"

_EGUsphere, 2025_

## Referee Comment (RC1)

**Overview**

This study presents a machine learning approach using OMPS-LP radiances to retrieve water vapor, with MLS water vapor profiles serving as the training target. The method employs a neural network (NN) model trained on 12-channel OMPS-LP inputs, and the manuscript outlines the application and evaluation of the resulting product.

Overall, this may be a valuable contribution, particularly in leveraging machine learning for satellite-based water vapor retrieval. However, I am concerned that the NN is not actually retrieving water vapor since the OMPS-LP instrument spectral resolution is too broad to capture the NIR water vapor lines in the 0.95µm region (Figure A, below). The authors need to prove that the system retrieves water vapor. In addition, several aspects of the study require clarification or improvement.

**Specific comment on retrieval**

SAGE III/ISS uses the 930 to 950 nm channels to retrieve water vapor. The SAGE spectrometer has a spectral resolution of ~3 nm (Davis et al. 2021). The SAGE water vapor retrieval is very noisy even with the high spectral resolution and solar radiation source. OMPS-LP uses scattered radiation, and spectral resolution is ~40 nm in this wavelength region. Looking at Figure A below, I have a hard time seeing how OMPS-LP can detect water vapor at all unless it is highly elevated (e.g. manuscript Figure 1).

Figure A. NIR water vapor line intensity (lower figure is a blow up of the upper figure). Blue boxes show the OMPS spectral resolution and yellow box shows the SAGE spectral resolution for comparison. Given the narrowness of the lines and spectral width, the changes in water vapor will be difficult to detect.

At background levels (~ 4.5 ppmv), the authors are getting ~0.25 ppmv variation (Fig. 6b). Given the wide spectral resolution of OMPS the channels 929, 943, 956, 970, and 983 reported on line 73. Is it possible that the NN is training on other factors and creating a water vapor simulation, because water vapor in the stratosphere is highly influenced by dynamics which is part of the NN training data. To truly demonstrate that the NN is retrieving water I suggest an experiment: fix the water vapor concentration to climatology. Then run the NN using the radiance variations in the other bands also inputting temperature and pressure. I suspect you will get the results shown in Fig 6 up to Hunga even though water vapor is not varying.

I would also like to see a regression plot where the variation in the water vapor at the different levels is regressed against the various bands. This is a kind of standard step in feature engineering for machine learning. I suspect you will find the highest correlation between temperature and pressure and the other bands are contributing little. This should tell us if the NN is actually using water band variations.

**Detailed comments and questions for revision:**

**1. Channel Selection and Feature Engineering**

Figure 1 appears to demonstrate this, but it lacks a legend explaining the color code for the weighting functions. It looks like the weighting functions (dln(I)/Dln(H2O) are near zero before Hunga so it isn't surprising that the factor increase will be large.

Given the potential for varying sensitivity, why not perform feature selection or regression analysis to identify the most informative channels? Why is it necessary to use 12 OMPS channels as inputs? What happens if you fix the temperature and pressure?

**2. Model Evaluation by Latitude**

I recommend including performance metrics (e.g., RMSE, bias) as a function of latitude, which may also capture dependence on solar zenith angle, given its inclusion in the input dataset.

**3. Ensemble Model Clarification**

You mention determining ensemble size based on prediction stability. Is the ensemble size consistent across all profiles, or determined dynamically?

What differentiates each ensemble member, like architecture, initialization, or hyperparameters?

**4. Normalization and RMSE Interpretation**

What are the units in Figure 2? Does Figure 2 use absolute RMSE? Variables are normalized in each altitude, an absolute value may misrepresent performance. Consider plotting relative RMSE (e.g., RMSE divided by median water vapor at each altitude) to better contextualize errors, especially at lower altitudes (<15 km) where water vapor concentrations are naturally higher. This would also help clarify if the elevated RMSE near the surface is a true error or a reflection of larger absolute values.

5. The statement "errors increase below 18.5 km..." needs clarification. Do you mean that measurement density is higher in the troposphere, or that variability increases? Does the sample size vary significantly with altitude?

**6. Concerns About Temporal Coverage and Generalization**

For the year dependence, Section 4.1 lacks clarity. You mention omitting 2024–present (Line 179), but training data is stated to cover 2014–2024 (Line 85). Did you use 2025 data? What is the exact time period excluded, and how does this affect inference quality?

Your explanation for 2024 being "special" is unconvincing – also see comment about Ruang above. The Hunga Tonga eruption occurred in early 2022, and the water vapor peaked shortly after. This does not justify 2024 as a critical component for training unless further supported by data.

**7. Feature Design and Model Limitations**

In your study, the year is not treated as an input feature. If year-to-year variation affects model performance, this could point to missing explanatory variables or insufficient feature engineering. You may consider a data imbalance or out-of-distribution (OOD) problem in your training.

8. In addition, given the relatively small number of input features except the 12 channels and model may be overfitting. I would like to see your support materials to make sure your model is not overfit.

Please consider revisiting the input space, especially if training struggles to generalize beyond 2024.

9. You state that model errors may not related to aerosol loading in Line 187. I am just curious like a time series of model errors alongside aerosol concentrations (e.g., before and after the 2022 eruption), do error patterns increase during high aerosol periods?

**10. Comparisons and Justification of External Datasets**

While comparisons with SAGE, ACE, and MLS are common, their measurement techniques differ significantly from OMPS-LP as you stated in the manuscript. This limits the interpretability of these comparisons. Since your model is trained on MLS water vapor, it makes most sense to validate primarily against MLS. In other words, the result shows differences, but these may stem from discrepancies between MLS and other datasets – see the MLS data quality and description document (Livesey et al., 2022), not from your model. The same remark can be applied to comparisons with M2-SCREAM.

**12. Figure 8.**

The claim that the NN methodology reduces drifts may be overstated. If the MLS data exhibits a decadal trend and your model was trained with shuffled input, it would be expected to replicate that trend. It does not make sense to me the model can do drift correction automatically. Please investigate and explain the reason for the difference before attributing it to NN drift correction.

**13. NOAA-21 Application (Section 4.7)**

While it's reasonable to apply the trained model to NOAA-21, the manuscript doesn't clearly justify the value of this step.

You acknowledge a bias/shift between SNPP and NOAA-21 radiances, which already limits comparability. The bias between two OMPS radiances obviously reflects in the inference. The statement in Line 290, suggesting the model may implicitly account for radiance bias, is likely overstated given the model's simplicity and data.

**14. Figures and Presentation**

Figure 1. Missing legend. Please indicate what each color represents.

Figure 6. Consider adding a third panel showing the difference between Figures 6a and 6b to better highlight anomalies or patterns not captured by direct comparison.

Figure 7. Since Figures 7a and 7b are expected to show similar results due to the consistent retrieval, they may be redundant. Consider removing 7a and 7b, and retain 7c, which provides more useful spatial comparison.

Line 180. The Ruang aerosols may have created problems in the April 2024 period

Line 207. Water vapor in the stratosphere doesn't have a diurnal cycle so why would time co-location make any difference unless the NN is using other gases such as O3 or temperature?

**Reference:**

Davis, S. M., et al. "Validation of SAGE III/ISS solar water vapor data with correlative satellite and balloon-borne measurements." *Journal of Geophysical Research: Atmospheres* 126.2 (2021): e2020JD033803.

---

## Author Comment (AC1)

We thank the anonymous reviewer for their thoughtful, detailed review of the manuscript, as it will improve the quality of the manuscript. Our response to each comment is provided below.

*Is it possible that the NN is training on other factors and creating a water vapor simulation, because water vapor in the stratosphere is highly influenced by dynamics which is part of the NN training data. To truly demonstrate that the NN is retrieving water I suggest an experiment: fix the water vapor concentration to climatology. Then run the NN using the radiance variations in the other bands also inputting temperature and pressure. I suspect you will get the results shown in Fig 6 up to Hunga even though water vapor is not varying.*

*I would also like to see a regression plot where the variation in the water vapor at the different levels is regressed against the various bands. This is a kind of standard step in feature engineering for machine learning. I suspect you will find the highest correlation between temperature and pressure and the other bands are contributing little. This should tell us if the NN is actually using water band variations.*

*1. Channel Selection and Feature Engineering*

*Figure 1 appears to demonstrate this, but it lacks a legend explaining the color code for the weighting functions. It looks like the weighting functions (dln(I)/Dln(H2O)) are near zero before Hunga so it isn't surprising that the factor increase will be large.*

*Given the potential for varying sensitivity, why not perform feature selection or regression analysis to identify the most informative channels? Why is it necessary to use 12 OMPS channels as inputs? What happens if you fix the temperature and pressure?*

We have updated Figure 1 to include the legend and adjusted the colors of each line to hopefully allow this to be better differentiated.

While a regression analysis is typical for feature engineering, it assumes that there is a linear relationship between the (possibly transformed) input-output pairs, but this is often insufficient for more complex problems, specifically problems that have complex non-linear relationships or highly correlated features, which is the case for this problem. Past work to use regression to determine a relationship between LP radiances and co-located $H_2O$ profiles was unsuccessful, indicating the relationship between LP radiances and $H_2O$ is more complex than can be captured by a simple regression analysis.

It is not strictly necessary to use 12 OMPS channels as inputs. In earlier stages, we also used ~50 channels, and while the results were similar, they were slightly worse than when we limited the number of channels. With straylight affecting LP's longer wavelengths, it's possible that the

additional wavelengths complicated the relationship and inhibited the NN from learning to properly account for that, but this is a minor effect considering the similarity in results. The important aspect is that a wide range of wavelengths is used to capture the spectral behaviors of different aerosol and scene reflectivity conditions, which enables the NN to differentiate these effects from $H_2O$.

However, we have performed several tests that seek to answer the feature importance question through other means.

Our initial model setup allowed for a simple test of perturbations in the temperature/pressure profiles. In March 2025, there was a switch in the LP ancillary product from using the GEOS FP-IT data to the new GEOS-IT product, which exhibited a discontinuity in the temperature data on the order of a few degrees Kelvin. If the temperature/pressure data were primarily driving the $H_2O$ predictions, then it would be expected to see differences between the model trained on the GEOS FP-IT temperature data but applied to the GEOS-IT temperature data, vs. a model trained exclusively on GEOS-IT data. We reapplied our methodology using the new GEOS-IT product throughout training and find our results for water vapor predictions in 2025 unchanged, indicating that the NNs are robust to small perturbations in temperature data.

We additionally investigated this question by training NNs without LP radiances or solar zenith angles, training NNs using climatological temperature/pressure profiles, and training on only LP radiances and solar zenith angles. When omitting LP radiance and solar zenith angles from training, we find that the model performs significantly worse, with larger root mean square errors and smaller $R^2$ values when applied to the test set (see Figure R1.1 below). The resulting tape recorder plot has worse agreement with the MLS tape recorder than what is presented in the manuscript, especially in the first weeks after the Hunga eruption as well as in 2025 (see Figure R1.2 below). Conversely, when using climatological temperature/pressure profiles, we find that the RMSE and $R^2$ values over the test set agree with those presented in the manuscript. Additionally, training on only the LP radiances also achieves similar RMSE and $R^2$ metrics as those presented in the manuscript. These results indicate that while the temperature/pressure data are useful, they are less important than the radiances when solving this problem.

[Figure]

**Figure R1.1.** Like Figure 2 in the manuscript, except additionally showing the performance metrics for a NN trained on only temperature and pressure data (dashed lines). The degraded performance in the stratosphere above 15-17 km suggests that the LP radiances provide important information that enables the determination of a better solution for retrieving stratospheric water vapor.

[Figure]

**Figure R1.2.** Like Figure 6 in the manuscript, except panel (a) shows results for the model trained only on temperature and pressure data.

*2. Model Evaluation by Latitude*

*I recommend including performance metrics (e.g., RMSE, bias) as a function of latitude, which may also capture dependence on solar zenith angle, given its inclusion in the input dataset.*

The percent bias between LP water vapor predictions and MLS per latitude is provided in the original manuscript; see Figure 4a.

We have added an additional panel to Figure 2 to show the relative RMSE and $R^2$ as a function of latitude as recommended. We find that the RMSE throughout the vast majority of the stratosphere is ~1/10 of the mean VMR. Below the tropopause, the RMSE is on the order of or larger than the mean VMR. For convenience, we provide that new panel below as Figure R1.3:

[Figure]

**Figure R1.3.** Plots of **(a)** relative RMSE and **(b)** $R^2$ as a function of latitude. The relative RMSE is shown on a logarithmic scale to better differentiate the transition in performance near the tropopause as well as minor variations in the stratosphere.

*3. Ensemble Model Clarification*

*You mention determining ensemble size based on prediction stability. Is the ensemble size consistent across all profiles, or determined dynamically?*

*What differentiates each ensemble member, like architecture, initialization, or hyperparameters?*

The ensemble size is constant across all profiles. As mentioned on lines 119-120, the architectures are identical among all ensemble members. Members are only differentiated by their random initialization. We have added additional text to better clarify this:

"Using the chosen architecture, we train an ensemble of 10 neural networks using a mean-squared-error loss function; **members are only differentiated by their random initialization**. **The size of the ensemble is held constant for all retrievals.**"

*4. Normalization and RMSE Interpretation*

*What are the units in Figure 2? Does Figure 2 use absolute RMSE? Variables are normalized in each altitude, an absolute value may misrepresent performance. Consider plotting relative RMSE (e.g., RMSE divided by median water vapor at each altitude) to better contextualize errors, especially at lower altitudes (<15 km) where water vapor concentrations are naturally higher. This would also help clarify if the elevated RMSE near the surface is a true error or a reflection of larger absolute values.*

Yes, Figure 2 shows the absolute RMSE, as indicated by the "VMR" units provided in the figure. However, it is a good point that this obfuscates how these RMSEs compare to the typical $H_2O$ VMRs at each altitude, and using a relative metric would better contextualize these errors. We have updated Fig. 2 to show the absolute RMSE divided by the training data set's average $H_2O$ VMR at each altitude, as suggested. We use the average rather than the median as the statistic had been previously calculated by the NN code.

*5. The statement "errors increase below 18.5 km..." needs clarification. Do you mean that measurement density is higher in the troposphere, or that variability increases? Does the sample size vary significantly with altitude?*

Yes, yes, and no, respectively.

The $H_2O$ VMR is significantly higher in the troposphere, and absolute errors are also larger in this region.

When considering percent differences between the LP predictions and co-located MLS profiles, the variability of these differences is larger in the troposphere; the differences are typically within 10% in the stratosphere with extreme differences of ~20%, while in the troposphere they can exceed 50%. We believe this may be due to a saturation effect, as the increased scattering in the upper troposphere likely limits the accuracy and precision of our measurements in this regime.

In Figure 2, the sample size is identical at all altitudes.

*6. Concerns About Temporal Coverage and Generalization*

*For the year dependence, Section 4.1 lacks clarity. You mention omitting 2024–present (Line 179), but training data is stated to cover 2014–2024 (Line 85). Did you use 2025 data? What is the exact time period excluded, and how does this affect inference quality?*

*Your explanation for 2024 being "special" is unconvincing – also see comment about Ruang above. The Hunga Tonga eruption occurred in early 2022, and the water vapor peaked shortly after. This does not justify 2024 as a critical component for training unless further supported by data.*

Lines 178-180, where we discuss omitting data from 2024-present, describes a separate experiment conducted.  The text has been updated to clarify this point:

> "**We carried out additional experiments where certain years were omitted from training and found that this** can be important for certain situations.  When omitting 2015-2016, …"

For the model we presented in the manuscript (the model that is currently producing the LP $H_2O$ products), training data covers 2014-2024 as described on line 85.  Thus, the only difference between our presented model and the model from the separate experiment is the inclusion of 2024 data during training.  2025 data are not used in training at any point.

As discussed in the manuscript, the exclusion of 2024 data impacts inference quality when applied to data from March 2024 and onward.  Note that this performance degradation is unrelated to Ruang, as Ruang did not erupt until mid-April 2024.  The explanation for this poor performance is shown in Figure 6: in the first half of 2024, the MLS tape recorder shows significantly elevated $H_2O$ above 30 km compared to the pre-Hunga period.  These conditions are not well represented in a 2014-2023 training data set (where 30+ km $H_2O$ enhancements are accompanied by different conditions than in 2024 and beyond), which leads to poor model generalization.  By including some of these data in training, model performance significantly improves in this regime, and it generalizes into 2025 where MLS also shows elevated $H_2O$ above 30 km.

*7. Feature Design and Model Limitations*

*In your study, the year is not treated as an input feature. If year-to-year variation affects model performance, this could point to missing explanatory variables or insufficient feature engineering. You may consider a data imbalance or out-of-distribution (OOD) problem in your training.*

Year is not treated as an input feature because year-to-year variability is implicitly contained within the LP radiances and, to a lesser extent, temperature/pressure data. Data imbalance was handled by subsampling the co-located data as discussed in lines 108-110, and the same lines also discuss a step taken to minimize the chances of an OOD problem. Despite that, it's possible that there is an OOD problem as the dimensionality of the problem (421 unique inputs) makes it difficult to truly determine this.

> *8. In addition, given the relatively small number of input features except the 12 channels and model may be overfitting. I would like to see your support materials to make sure your model is not overfit.*
>
> *Please consider revisiting the input space, especially if training struggles to generalize beyond 2024.*

There are 421 unique input features (1440 input features when including redundant inputs for the image-based processing used) mapped to 30 output features. This is not typically considered a small number of input features in the ML literature. The model is not overfit as evidenced by various performance metrics being similar on both the validation (occasionally seen during training) and test (never seen during training) data. See Figure R1.4 below, which is analogous to the manuscript's Figure 2 except showing the metrics for the validation set in addition to the test set. The test and validation curves are nearly identical, indicating that the model generalized to unseen data and did not overfit the training/validation data.

Additionally, we show below in Figure R1.5 the median differences between LP and MLS for various years. Focusing in on 2025, we can see that the differences above 25 km are consistent with the years considered during training. Below 25 km, the errors for 2025 are around -6-7%, whereas the years considered during training are typically within 2%. However, it is important to note that there are significantly fewer LP-MLS co-locations in 2025 due to the MLS duty cycling that reduced MLS observations to ~6 days per month, which may be affecting these comparisons. Nevertheless, the errors in this regime are generally less than the LP-ACE comparisons shown in the manuscript's Figure 3b and they are comparable to the LP-SAGE comparisons shown in the manuscript's Figure 3c. If the model were overfitted, it would be expected that the 2025 errors would be significantly different than the years seen during training.

[Figure]

**Figure R1.4.** Like Figure 2 in the original manuscript, except also including the performance metrics for the validation data. The lack of a performance gap between the validation and test set metrics indicates that the model has generalized well to unseen data.

[Figure]

**Figure R1.5.** Like the manuscript's Figure 3a, but showing individual years as well as the pre- and post-Hunga periods.

*9. You state that model errors may not related to aerosol loading in Line 187. I am just curious like a time series of model errors alongside aerosol concentrations (e.g., before and after the 2022 eruption), do error patterns increase during high aerosol periods?*

Figure R1.5 above shows that the post-Hunga period has a 1-2% difference compared to the pre-Hunga period. Figure R1.6 below shows the requested time series plot. Each vertical line shows the average percentage difference between LP-MLS co-locations for each LP orbit (LP has 14-15 orbits per day). Near the top of the plot, three large eruptions are marked (Calbuco, Hunga, and Ruang). While there are occasional orbits with larger errors than the average, they are not correlated with major eruptions, consistent with what was reported in the manuscript.

[Figure]

**Figure R1.6.** Time series plot of the average percentage difference between LP-MLS co-locations per LP orbit. Three major eruptions are denoted on the plot for context.

*10. Comparisons and Justification of External Datasets*

*While comparisons with SAGE, ACE, and MLS are common, their measurement techniques differ significantly from OMPS-LP as you stated in the manuscript. This limits the interpretability of these comparisons. Since your model is trained on MLS water vapor, it makes most sense to validate primarily against MLS. In other words, the result shows differences, but these may stem from discrepancies between MLS and other datasets – see the MLS data quality and description document (Livesey et al., 2022), not from your model. The same remark can be applied to comparisons with M2-SCREAM.*

Yes, we agree on this point.  The primary validations are with MLS, as that is what we trained on.  The purpose of including comparisons with additional instruments is to show that the model is more generally applicable, that is, it doesn't *only* work where LP is co-located with MLS.  The differences between LP and the other instruments are indeed a product of the discrepancies between MLS and those other datasets, as our NN approach mimics the MLS product (e.g., line 235-236, "… or MLS is biased low in this regime and our LP product has inherited this bias."), but it is an important element to show the generalization of the approach.

(Noting that there is no comment #11)

*12. Figure 8*

*The claim that the NN methodology reduces drifts may be overstated. If the MLS data exhibits a decadal trend and your model was trained with shuffled input, it would be expected to replicate that trend. It does not make sense to me the model can do drift correction automatically. Please investigate and explain the reason for the difference before attributing it to NN drift correction.*

It is a good point raised here and by the other reviewer that the presented results do not support a conclusion of the NN model reduces MLS drifts, as we did not compare trends in water vapor derived from LP and MLS with more accurate frost point measurements.  We have revised the text to remove mentions of drift reduction and instead focus on the presented trends:

"Regarding water vapor trends, the LP product generally shows  weaker trends in the stratosphere when compared with MLS.  **In some locations (particularly south and southeast Asia, central Africa, and central America), the LP product shows greater trends in the upper troposphere.**  Where both products show …"

The NN is attempting to minimize the mean squared error across all training and validation cases, so the differences in trends are a product of that process. Presumably, the difference in trends is related to differences in instrument performance over time between LP and MLS, that is, the sensors will not degrade in the same way, but it is beyond the scope of this work to definitively determine this.

*13. NOAA-21 Application (Section 4.7)*

*While it's reasonable to apply the trained model to NOAA-21, the manuscript doesn't clearly justify the value of this step.*

*You acknowledge a bias/shift between SNPP and NOAA-21 radiances, which already limits comparability. The bias between two OMPS radiances obviously reflects in the inference. The statement in Line 290, suggesting the model may implicitly account for radiance bias, is likely overstated given the model's simplicity and data.*

The justification for this step is given at the end of Section 3.3:

"Finally, given that OMPS LP is onboard the NOAA-21 satellite and planned to launch onboard two additional satellites in the coming years, we apply our SNPP-trained model to NOAA-21 OMPS LP measurements to determine whether our model can generalize to future iterations of the same instrument …"

We have revised the text to more clearly explain why this is important:

"NOAA-21 OMPS LP has an insufficient period overlapping with MLS measurements (2023 – present, with MLS only taking measurements for ~6 days per month since May 2024), inhibiting the use of NOAA-21 OMPS LP data for our NN training methodology. Given the imminent termination of Aura MLS in two years and that the SNPP satellite will presumably cease operations before the end of NOAA-21 or the subsequent JPSS satellites, we test the application of our SNPP-trained model to NOAA-21 OMPS LP measurements to determine whether our model can generalize to future iterations of the same instrument …"

*14. Figures and Presentation*

*Figure 1. Missing legend. Please indicate what each color represents.*

Done

*Figure 6. Consider adding a third panel showing the di^erence between Figures 6a and 6b to better highlight anomalies or patterns not captured by direct comparison.*

Done

*Figure 7. Since Figures 7a and 7b are expected to show similar results due to the consistent retrieval, they may be redundant. Consider removing 7a and 7b, and retain 7c, which provides more useful spatial comparison.*

We appreciate the suggestion, but we think that it is important to include both panels (a) and (b) to highlight the close agreement between LP and MLS.

*Line 180. The Ruang aerosols may have created problems in the April 2024 period*

As discussed above, Ruang erupted in mid-April 2024, but the issues emerged in March 2024, indicating they are unrelated to the Ruang eruption.

*Line 207. Water vapor in the stratosphere doesn't have a diurnal cycle so why would time co-location make any difference unless the NN is using other gases such as O3 or temperature?*

The NN does indeed rely on temperature in part, as shown above in comment #1.  In the lower stratosphere, dynamics drive changes in trace gas concentrations; differences of several hours between measurements can lead to LP and MLS measuring different air masses with different concentrations of $H_2O$, which could limit the accuracy of the trained model.  However, even when isolating this variable, the number of co-locations is a main limiting factor, as found by our experiment where we restrict the data set size for MLS co-locations (lines 208-210).